


**Quantitative Study of Storm Surge Risk Assessment in Undeveloped Coastal Area of China**
**Based on Deep Learning and Geographic Information System (GIS) Techniques: A Case**
**Study of Double-Moon Bay Zone**
Lichen Yu [a, d], Shining Huang [c], Hao Qin [*, a, d], Wei Wei [a, d], Lin Mu [*, b]
[a] Hubei Key Laboratory of Marine Geological Resources, College of Marine Science and
Technology, China University of Geosciences, Wuhan, China, 430074
[b] College of Life Sciences and Oceanography, Shenzhen University, Shenzhen, China, 518060
[c] Marine Information Center, Department of Natural Resources of Huizhou Bureau, Huizhou,
China, 516003
[d] Shenzhen Research Institute, China University of Geosciences, Shenzhen, China, 518057
[*] Corresponding authors: Hao Qin (qh1qh100@alumni.sjtu.edu.cn); Lin Mu (moulin1977@h
otmail.com).
**Abstract**
Storm surge is a common nature disaster in China southern coastal area, which usually causes
heavy human life and economic losses. With the economic development and population
concentration of coastal cities, the storm surges may result in more impacts and damage in the
future. Therefore, it is of vital importance to conduct risk assessment to identify high-risk areas
and evaluate economic losses. However, quantitative study of storm surge risk assessment in
undeveloped areas of China is difficult, since there is a lack of building characters and damage
assessment data. Aiming at the problem of data missing in undeveloped areas of China, this paper
proposes a methodology for conducting storm surge risk assessment quantitatively based on deep
learning and geographic information system (GIS) techniques. Five defined storm surge
inundation scenarios with different typhoon return periods are simulated by coupled
FVCOM-SWAN model, the reliability of which is validated using official measurements. Building
footprints of the study area are extracted through TransUNet deep learning model and Remote
Sensing Image (RSI), while building heights are obtained through Unmanned Aerial Vehicle (UAV)
measurement. Subsequently, economic losses are quantitatively calculated by combing the
adjusted depth-damage functions and overlay analysis of the buildings exposed to storm surge
inundation. Zonation maps of the study area are illustrated to display the risk levels according to
the economic losses. The quantitative risk assessment and zonation maps can help the government
to make storm surge disaster prevention measures and optimize land use planning, and thus to
reduce the potential economic losses of the coastal area.

**Keywords:** Strom surge; Quantitative risk assessment; GIS; Deep learning; Risk zonation map

**1. Introduction**
Storm surge, defined as the abnormal rise of water over and above the normal astronomical
tide, and is expressed in terms of height above predicted or expected tide levels. Mostly, the surge
is generated by a strong atmospheric disturbance, and it becomes particularly catastrophic when it
happens to coincide with an astronomical high tide. In that case, the surge-driven coastal flooding
may inundate buildings and cropland, cause significant casualties and economic losses. Storm
surges have caused widespread damage worldwide. In 2013, super typhoon Yolanda as the worst




typhoon in last 30 years, pounded the Philippines. It caused 6293 individuals reported dead, 28689
injuries and 1061 individuals missing, with estimated damages totaling 864 million US dollars
(Mcpherson, 2015). Hurricane Harvey struck Texas in August 2017, resulting in approximately
100 deaths and economic losses exceeding 125 billion dollars (Lee, 2021). In China, storm surges
also pose a frequent threat in the coastal cities. In the last decade, China has experienced an
average of 8.5 storm surge disasters annually, with an average damage amount of 6815.8 million
yuan per year, where Guangdong and Zhejiang Provinces are the most affected coastal areas
(China Marine disaster bulletin, 2022). For example, Typhoon Hato in 2017, Typhoon Mangkhut
in 2018, Typhoon Lekima in 2019 has caused significant damage to coastal cities in China, and
resulted great losses of life and property. For the past few years, as the rapid development of
population and economic in China coastal area, the potential monetary loss grows accordingly
(Fang et al., 2021; Ji et al., 2020; Mcgranahan et al., 2007; Seto et al., 2011). Therefore, it is
crucial to implement risk assessment and mapping strategies to effectively reduce these risks and
mitigate the impact of storm surges.
Storm surge hazard assessment is an essential component of storm surge risk assessment and
zoning, aiming to evaluate the hazard intensity of disasters, through numerical simulation of storm
surge processes, estimation of storm surge for selected return periods, and computation of the
probable maximum storm surge (Shi et al., 2013). Therefore, the numerical simulation of storm
surge is a key step for storm surge risk assessment. However, because of the limitation of
historical storms and the nondeterminacy of future storm, numerical simulation of storm surges is
usually used to determine storm levels. Advanced Circulation Mode (ADCIRC) is a widely used
hydrodynamic model in coastal area. For example, Vijayan et al. (2021) utilized ADCIRC model
to simulate storm surges and tides during the hurricane that land on Florida in 2018, for the
purpose of comparing the different impact of wind model Holland 1980 and Holland 2010. Wang
et al. (2021a) and Liu and Huang (2020) used ADCIRC and Simulating Waves Nearshore (SWAN)
coupled model respectively simulate the storm surge and wave in the sea near Shandong Peninsula
and Taiwan, and the hazard assessment and model verification were carried out respectively.
Delft3D is a comprehensive numerical modeling system for simulating hydrodynamic processes.
Hu et al. (2022) adapted a pre-validated Delft3D-based hydrodynamic model proved the impact of
levee opening at selected locations was minor. Lyddon et al. (2019) used Delft3D-FLOW-WAVE
model calculate the tide and wave in the Severn Estuary, the result pointed out the importance of
locally generating winds in simulation of water level and wave height. Finite Volume Coastal
Ocean Model (FVCOM) is another widely used numerical model for simulating hydrodynamic
processes. Zhang et al. (2020) conducted a series of modeling experiments with the purpose of
assessing the impact of storm and evaluated the flood protection by using FVCOM inundation
model. Zhu et al. (2022) realized WRF-SWAN-FVCOM coupling simulation to analyze the
spatial-temporal evolution laws, and the result demonstrate the method can predict hydroelastic
responses of the maritime airport under the impact of typhoons, currents and waves.
It has been demonstrated that it is critical to include tide and sea-water-level variations in
shelf and nearshore wave simulations (Masson, 1996). Furthermore, the sea water level could be
significantly affected by strong tide and typhoon-induced wind in complex coastal seas and then
modulate the wave properties (Yang et al., 2020). Coupled FVCOM-SWAN model, with the
foundation of FVCOM's finite volume method, unstructured grid, and adaptable boundary
condition handling capability, integrating the hydrodynamic and wave processes of SWAN,


possesses the ability to provide simulation result more quickly and accurately. In this circumstance,
coupled FVCOM-SWAN model is used in this research for simulating the inundation of storm
surge.
Coastal risk assessment can be categorized into two primary classifications: qualitative and
quantitative. In the realm of qualitative assessment, entropy weight method, Analytic Hierarchy
Process (AHP) and other methods are widely used. Ramkar and Yadav (2021) used AHP in
combination with Geographic Information Systems (GIS) for proposing a flood risk map, which
can identify the high-risk areas efficiently. Malekinezhad et al. (2021) combined the entropy
weight method and GIS, and conducted a flood vulnerability analysis for Hamadan city. The result
highlighted the advantages of entropy weight method comparing to normal spatial overlay method.
Besides, Pathan et al. (2022) and Rafiei-Sardooi et al. (2021) made use of Technique for Order
Preference of Similarity by Ideal Solution (TOPSIS). The former pointed out the advancement of
TOPSIS by comparing with AHP, and the latter combined machine learning and TOPSIS to
analyze urban flood vulnerability. Unlike qualitative risk assessment, quantitative risk assessment
enables the quantification of damages and risks in monetary terms. The most commonly used
approach to assess direct damages is based on depth-damage curves (De Moel and Aerts, 2011;
Merz et al., 2007; Smith, 1994). Thieken et al. (2008) presented the Flood Loss Estimation Model
for the private sector (FLEMOps) through using the Germany flood losses data in August 2002,
and the group further established model for commercial sector in 2010 (Kreibich et al., 2010).
Zhai et al. (2005) derived multi-factor loss functions for buildings in Nagoya, Japan using
empirical data from Tokai flood in 2000, and Grahn and Nyberg (2014) established functional
relationships utilizing the house insurance claims data caused by lake flooding. Except for
buildings, Yazdi and Salehi Neyshabouri (2012) and Hess and Morris (1988) respectively built
several uni-variable functions and multi-factor functions for kinds of crops and grassland. In
recent years, machine learning is also introduced in quantitative loss assessment, for example,
Merz et al. (2013) developed a tree-based approach using Regression Tree and Bagging
Regression Tree as machine-learning methods to analysis of direct building damage to private
homes. Paprotny et al. (2020) proposed a Bayesian Network damage model (a
Supervised-Machine-Learning method), and reached a good accuracy of predictions of building
losses.
The essence of quantitative risk assessment lies in analyzing the interaction between
exposure factors and hazards (Adnan et al., 2020; Armenakis and Nirupama, 2013; Granger, 2003;
Kron, 2005), therefore it's crucial to quantify the direct tangible damage of elements at risk.
Buildings are important exposure elements, as they are the gathering place of population and
property. Building footprint data is necessary for evaluating the vulnerabilities of a building, as it
provides essential information about the buildings, including spatial location, distribution, and
boundaries and so on (Mharzi Alaoui et al., 2022). It's also of great significance in risk assessment,
primarily due to its ability to identify high-risk areas, assess building vulnerability and estimate
potential damage (Gacu et al., 2023; Wu et al., 2019). Extracting building footprints from remote
sensing images has been widely used in many fields. For example, in urban planning, Zhou et al.
(2004) used building footprint data and LiDAR point cloud data for urban 3D modeling; Tang et al.
(2006) proposed a GIS-based landscape index combing with remote sensing to analyze urban
sprawl spatial fragmentation. In disaster management, Liu et al. respectively evaluated seismic
vulnerability in Urumqi and Weinan in China (Liu et al., 2019; Liu et al., 2020). In navigation,





Rousell and Zipf (2017) proposed a prototype navigation service based on multi-index in OSM
dataset and building footprints, and Chen and Gao (2019) merged GPS pseudorange, LiDAR
odometry measurements and building footprint to offer a UAV navigation algorithms. However,
there is a lack of building footprints extraction and application in the realm of storm surge
assessment.
In view of the aforementioned information, regarding storm surge qualitative risk assessment,
there is a stringent requirement for both the quality and timeliness of land use data, which means
that the risk assessment cannot be generated in real time, and the qualitative risk assessment also
can't evaluate the risk level through the intuitive value of economic loss. In the realm of
quantitative risk assessment, building a uni-variable or multi-factor empirical model requires
complete and substantial data, and the published models generally only provide uni-variable
functions ignoring the building height as a factor, or have regional limitations. Additionally, for
the coastal regions of China, which are often affected by storm surge disasters, they tend to have
relatively low levels of economic development. Under the circumstances, the data needed to
conduct flood risk assessment is generally in a state of absence.
In response to the challenges mentioned above, the scientific goal of this paper is to propose a
quantitative storm surge risk assessment method for underdeveloped areas based on deep learning
and GIS techniques. First, on the basis of high-resolution DEM and seawall data measurement,
five defined storm surge inundation scenarios with different typhoon return periods are simulated
by employing the coupled FVCOM-SWAN model. Subsequently, TransUNet is introduced as a
deep learning method to extract building footprint, and building's height data is acquired through
UAV measurement. Since data on relevant disaster losses in underdeveloped regions are lacking,
empirical modeling was deemed impractical. Therefore, the adjustment of the JRC's
depth-damage curves by the HAZUS is chosen to take the impact building's height into
consideration, thus to conduct a quantitative assessment with more accuracy. Finally, combining
hazard map, exposure elements and adjusted depth-damage curves, the quantitative risk zoning
maps are conducted. The risk zoning maps can assist decision-makers in identifying high-risk
sub-zones and planning disaster prevention measures. Accordingly, the novelty can be seen in
obtaining refined exposure elements data through deep learning and UAV, addressing the lack of
historical storm surge economic loss data and considering the effect of building height on
economic loss through the adjustment of existing depth-damage curves.
**2. Study area and datasets**
*2.1. Study area*
Being the shipping hub in the South China Sea, Guangdong province, located in southern
China, has become the largest economic province in China since 1989, with a GDP of 129118.6
billion yuan in 2022. Due to the seaborne trade, Guangdong has been the largest economic
province in China since 1989, which reached a GDP of 129118.6 billion yuan in 2022. However,
just as mentioned above, Guangdong is relatively vulnerable to storm surges because of its
geographical characteristics, such as Typhoon Hato and Typhoon Mangkhut.
Huizhou is one of the cities in Guangdong province, and also one of the central cities of Pearl
River Delta region. It's located at on the east coast of Guangdong-Hong Kong-Macao Greater Bay
Area, the GDP reached 540.1 billion yuan in 2022, with the highest growth rate in Guangdong.
Pinghai Town located at the southernmost of Huizhou, and has a registered population of about
forty thousand. Its economic source mainly depends on various crops and seafood products. Due


to its coastal geographical characteristics and the presence of Pinghai Ancient City, the town has
become a cultural tourist destination and can therefore be defined as a cultural tourist town.
In this paper, the chosen study region is the coastal area of Pinghai town, named the
Double-Moon Bay Zone. It covers ten villages in total, including Foyuan, Dayuan, Yuye, Xinliao,
Xin village, Shazuiwei, Cajia, Nanshe, Daao, and Harbor community. These years, the region has
been developed in tourism and real estate project development including construction of hotel,
resort, and high-end business district, which vastly prompt the financial development. It is
foreseeable that the population and economy of the region will growth rapidly. However, the
region's general economic status, which remains relatively low, and it consequently gives rise to
the challenge of data scarcity and limited accessibility. In addition, the region is susceptibly
affected by the tropical cyclones during the season running from April to November (Wang et al.,
2021b). Recent years, more than ten typhoons have affected the study area, including Typhoon
Lekima, Typhoon Haishen, Typhoon Kanuni etc. The general location and information about the
study area is shown on Fig. 1.

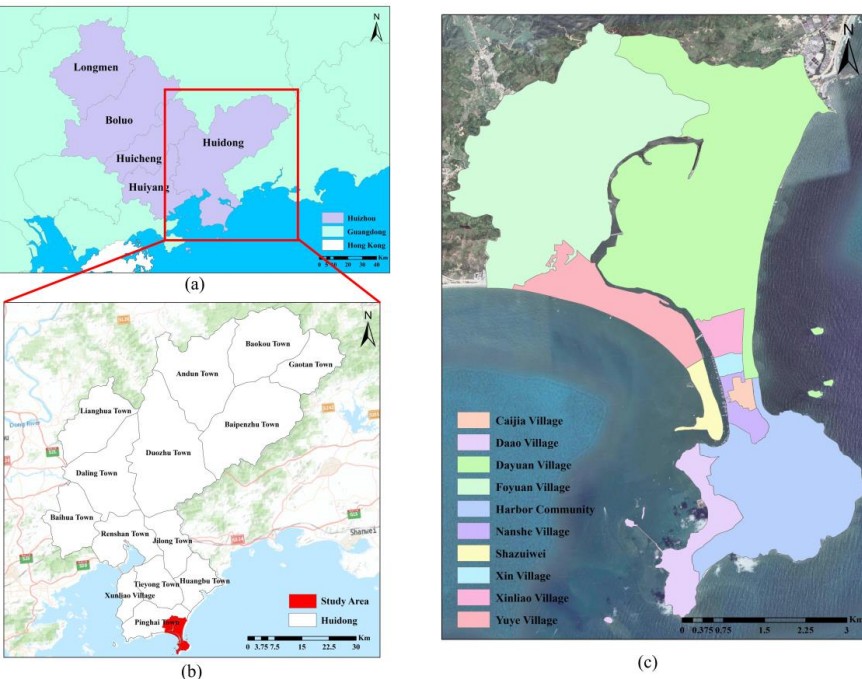

**Fig. 1.** The maps of locations in the study: (a) The map of Huizhou; (b) The map of study area in
Huidong, the base map is obtained from ESRI; (c) The village map of study area, the base map is
obtained from © GoogleMaps (map data © 2023 Google).

*2.2. Data source*
In order to accomplish the research, the data used is obtained from various sources, here is
the describe of different data:
(1) Land Cover Types data: the data is obtained from the Department of Natural Resources of
Huizhou Bureau. It contains multiple land cover types including forest, cropland, residential land,
etc. It is used to calculate vulnerability level.





(2) Remote sensing images: the remote sensing images are obtained from Chang Guang
Jilin-1 satellite. Chang Guang Satellite technology CO., LTD was founded on December 1st 2014,
which is the first and the largest commercial satellite corporation in China. Jilin-1 is the first
self-developed commercial high-resolution satellite. The images from Jilin-1 satellite have a
resolution of 50 cm, and have five spectral channels: Panchromatic band; Blue band; Red band;
Green band; Near Infrared band. The images consisting of blue band, red band, green band are
utilized to combine deep learning method, thus achieve the extraction of buildings.
(3) Unmanned Aerial Vehicle (UAV) data: the UAV data is generated by oblique photography,
and is organized by Open Scene Graph Binary format. The UAV data is obtained from Department
of Natural Resources of Huizhou Bureau, and the data is utilized for buildings' height calculation.
(4) Digital elevation model (DEM) data: the DEM data is captured by manual observation in
2018, with the resolution of 0.3m. The coordinate system and file organization are originally
CGCS 2000 and txt file, and further transformed to WGS 1984 and raster format to make use of
these data in the research. The data contains the elevation information for the study region.
Besides, the seawall data is also obtained manually. Both data are used in modeling of storm
surges for simulating the hazard maps.
(5) ERA5 data: ERA5 is the fifth generation of the European Reanalysis dataset produced by
the European Centre for Medium-Range Weather Forecasts (ECMWF), and it provides the
comprehensive and high-resolution atmospheric and climate data. In this study, the data is used in
conjunction with the Holland method to generate fused wind field data, which is subsequently
utilized for storm surge simulations.
(6) Historical typhoon data: the historical typhoon data including typhoon track, typhoon
pressure, and velocity are obtained through China Meteorological Administration Typhoon
Network Website. The historical data is employed to assess the reliability and validity of the
model.
(7) Administrative Boundary data: the data is obtained from National geographic information
public service platform, and it contains administrative boundaries at village level. There are ten
villages in the study area.

## 3. Method

The methods in this study aim to assess quantitative direct tangible damage over the study
area consists of following steps: hazard assessment; exposure assessment; vulnerability
assessment; risk assessment, and the flowchart of the procedure is illustrated in Fig. 2.
First, with respect to hazard assessment, five storm surge scenarios are defined. After
constructing wind field through Holland model, the inundation area and depth of different typhoon
return periods are simulated by utilizing the coupled FVCOM-SWAN model. In exposure
assessment, building footprints and heights are extracted by introducing a deep learning method
TransUNet and shadow calculation. Then the hazard maps are overlaid to identify the elements at
risk. Considering the effect of building's floor in flood monetary loss estimation, the JRC's
depth-damage functions are adapted representing the vulnerability of different exposed elements.
Eventually, the economic loss of different typhoon scenarios can be summarized and the risk
assessment is conducted through multiplying the temporal probability. Moreover, the quantitative
zoning maps of four risk levels are generated through zonal statistic.

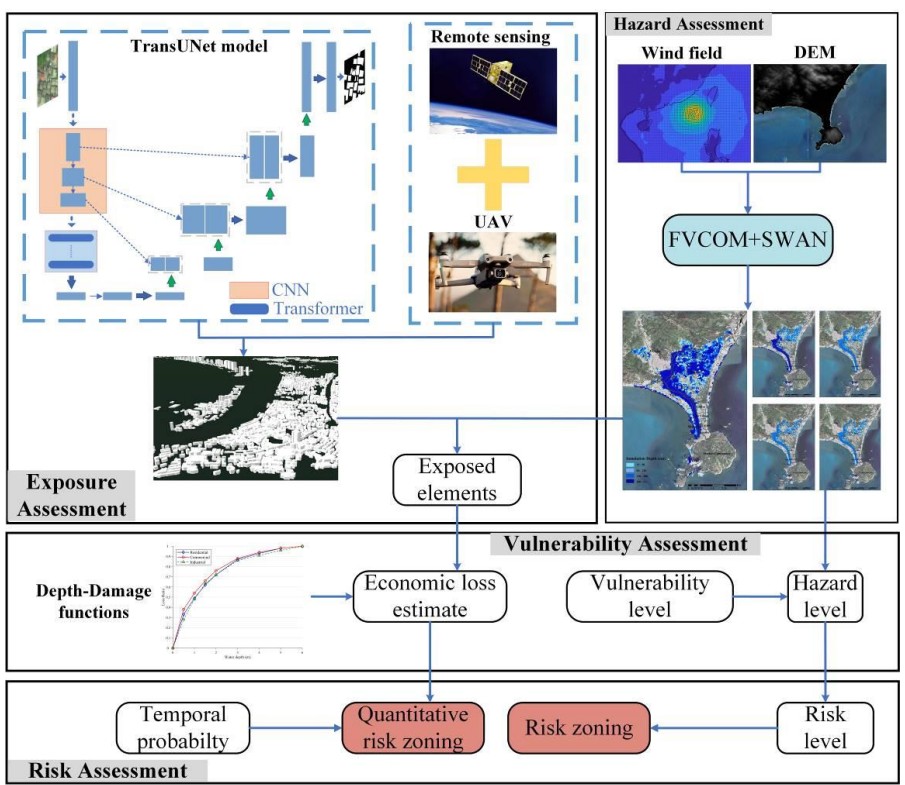

**Fig. 2.** The flowchart of the presented storm surge quantitative risk assessment method. The base map is obtained from © GoogleMaps (map data © 2023 Google).

*3.1. Strom surge inundation simulation*

Finite Volume Coastal Ocean Model (FVCOM), is a coastal ocean circulation model, which was originally developed by Chen et al. (2003), and further improved by the University of Massachusetts and the Woods Hole Oceanographic Institution. The following are the governing equations of FVCOM, comprising momentum, continuity, temperature, salinity, and density equations:

$$\frac{\partial u}{\partial t} + u\frac{\partial u}{\partial x} + v\frac{\partial u}{\partial y} + wfv = -\frac{1}{\rho_o}\frac{\partial P}{\partial x} + \frac{\partial}{\partial z}\left(K_m\frac{\partial u}{\partial z}\right) + F_u \tag{3.1}$$

$$\frac{\partial v}{\partial t} + u\frac{\partial v}{\partial x} + v\frac{\partial v}{\partial y} + w\frac{\partial v}{\partial z} + fu = -\frac{1}{\rho_o}\frac{\partial P}{\partial y} + \frac{\partial}{\partial z}\left(K_m\frac{\partial v}{\partial z}\right) + F_v \tag{3.2}$$

$$\frac{\partial P}{\partial z} = -\rho g \tag{3.3}$$

$$\frac{\partial u}{\partial x} + \frac{\partial v}{\partial y} + \frac{\partial v}{\partial z} = 0 \tag{3.4}$$

$$\frac{\partial T}{\partial t} + u\frac{\partial T}{\partial x} + v\frac{\partial T}{\partial y} + w\frac{\partial T}{\partial z} = \frac{\partial}{\partial z}\left(K_h\frac{\partial T}{\partial z}\right) + F_T \tag{3.5}$$





$$\frac{\partial S}{\partial t} + u\frac{\partial S}{\partial x} + v\frac{\partial S}{\partial y} + w\frac{\partial S}{\partial z} = \frac{\partial}{\partial z}\left(K_h\frac{\partial S}{\partial z}\right) + F_S \tag{3.6}$$

$$\rho = \rho(T,S) \tag{3.7}$$

Where $x$, $y$ and $z$ respectively represent the east, north and vertical coordinate axes in the
Cartesian coordinate system; $u$, $v$ and $w$ are the velocity components in $x$, $y$,
$z$ directions; $T$, $S$ and $\rho$ are the temperature, salinity and density; $P$ is the pressure and
$f$ stands for the Coriolis parameter; $K_m$ is the vertical eddy viscosity coefficient and $K_h$ is
the vertical eddy diffusivity coefficient for heat; $g$ is the gravitational acceleration; $F_u$, $F_v$,
$F_T$, and $F_S$ are the horizontal diffusion terms.
Simulating Waves Nearshore (SWAN) is the third-generation offshore wave model developed
by Delft University of Technology and it was originally proposed by Booij et al. (1996). The
governing equation of the model is shown as

$$\frac{\partial}{\partial t}N + \frac{\partial}{\partial x}C_xN + \frac{\partial}{\partial y}C_yN + \frac{\partial}{\partial \gamma}C_\gamma N + \frac{\partial}{\partial \theta}C_\theta N = \frac{S}{\gamma} \tag{3.8}$$

Where $N$ is the wave action density; $\theta$ is the propagation direction; $C_x$, $C_y$ are
respectively the $x$, $y$ components of propagation speed and $C_\gamma$, $C_\theta$ are the $\gamma$,
$\theta$ components of propagation cospeed; $\gamma$ and $S$ respectively represent the frequency and the
source term for the wave energy.
FVCOM and SWAN both use the unstructured triangular grid to subdivide the South China
Sea, and the latitude and longitude range of the region is 13°N - 29°N, 109°E-122°E. The SWAN
parameters are set as follows: wind input growth term and whitecap dissipation term are the
Komen scheme; Bottom friction dissipation is parameterized using the Madsen vortex viscosity
model; The nonlinear interactions are implemented using three-wave and four-wave nonlinear
interaction schemes. The input wind field is the fusing wind field derived from ERA5 and the
Holland method. The open boundary forced tidal elevation of FVCOM is conducted by calculating
the harmonic constants for the eleven major astronomical tidal constituents, namely M2, N2, S2,
K2, K1, O1, P1, Q1, MS4, M4, and M6. The forcing field is the fusing wind field and the wave
data generated by SWAN. The external model time step for the model is set to 0.75 second, while
the internal model time step is set to 7 seconds.
In the present study, FVCOM-SWAN coupling method is utilized for simulating the
inundation caused by storm surge. Specifically, following the modification of typhoon Mangkhut's
central pressure, velocity, and track data, the data is utilized as input for the Holland typhoon wind
field model, subsequently yielding the wind field outcome. The wind field data extracted is fed
into the SWAN model to generate wave data. Then, both the wind data and wave data are input
into the FVCOM model to calculate the extent of inundation.

### 3.2. Buildings extraction

The deep learning model used in the research is TransUNet (Chen et al., 2021), which was
originally proposed for medical images segmentation. TransUNet incorporates transformer in
encoder within the architecture of U-shape network, consequently makes use of the advantage of
global information extraction while fusing the superficial and deep features. On the mission of
building extraction, the target is to segment the building's area precisely. The TransUNet model
can effectively identify the boundary between buildings and background, which enables the model
to be competent for extracting the buildings in different size and shape.


The following is relevant introduction of the structure of the model.
*3.2.1. Transformer in TransUNet*
Transformer was first proposed by Sutskever et al. (2014), which was originally utilized for
machine translation. However, as more variants of transform were developed, people found
transform also perform well in multiple tasks, such as natural language processing (NLP),
computer vision (CV) and automatic speech recognition (ASR).
The transformer encoder is composed of L layers of Multi-head Self-Attention (MSA), Layer
normalization (LN) and Multi-Layer Perceptron (MLP), the structure is shown in the Fig. 3(a), and
the equations of Query-Key-Value (QKV) self-attention and MSA are shown below:

$$\text{Attention}(Q,K,V) = \text{softmax}(\frac{QK^{\text{T}}}{\sqrt{D_k}})V \tag{3.9}$$

$$\text{MultiHeadAttn}(Q,K,V) = \text{Concat}(\text{head}_1,...,\text{head}_H)\mathbf{W}^O \tag{3.10}$$

$$\text{head}_i = \text{Attention}(QW_i^Q, KW_i^K, VW_i^V) \tag{3.11}$$

Where $Q$, $K$, $V$ are respectively the Query, Key, Value vector. $\sqrt{D_k}$ is the scaled dot-
-product attention. $\mathbf{W}^O$, $W_i^Q$, $W_i^K$, $W_i^V$ are respectively the corresponding linear mapping, which
convert $Q$, $K$, $V$ and the output to the specified dimension.
The MSA has a positive effect on helping the model identify the target objects and
background, thus the neutral network can learn more information form the target. LN is deemed to
stabilize the deep network training, which can prevent unstable gradient, model degradation, etc.
The module receives the 2d flatted patches from the image's patches. Due to it is different from
CNN or RNN, apart from map the vectorized patches to D-dimensional embedding space,
transformer needs to apply additional position encoding for retaining the patch's positional
information.
*3.2.2. Structure of TransUNet*
The overall structure of TransUNet is reference to U-Net, which is a U-shaped
Encoder-Decoder structure, and the structure diagram is shown in the Fig. 3(b).
Encoder: the origin image is put into the CNN part for feature extraction, after the processing
of position encoding and flatten, the patches are further put into the transformer module. The
transformer module consists of 12 transformer layers. The CNN part is implemented through
using resnet50, which include 3 blocks in total, and each block output the hidden feature for skip
connection.
Decoder: reshape the output sequence from encoder and then cascade up-sampling after
transforming the number of channels. During the process, the skip connection is introduced by
using the feature map hereinbefore. In the end, the segmentation result is generated.
In conclusion, TransUNet is the combination of U-Net and transformer, which is designed to
make use of the advantage from both structures. The Global Attention from transformer can
contribute to learn the global information, while the skip connection from U-shape network can
contribute to get more information from shallow feature map output from CNN, and also CNN
performs better in extracting the local information. In this research, buildings images are similar to
medical images, with the features like high complexity level, large range of gray values. The skip
connection structure can simultaneously acquisition of low-level semantic features and high-level
semantic features, and transformer can conduce identify the buildings from background, thus
TransUNet achieves a high accuracy in buildings segmentation.

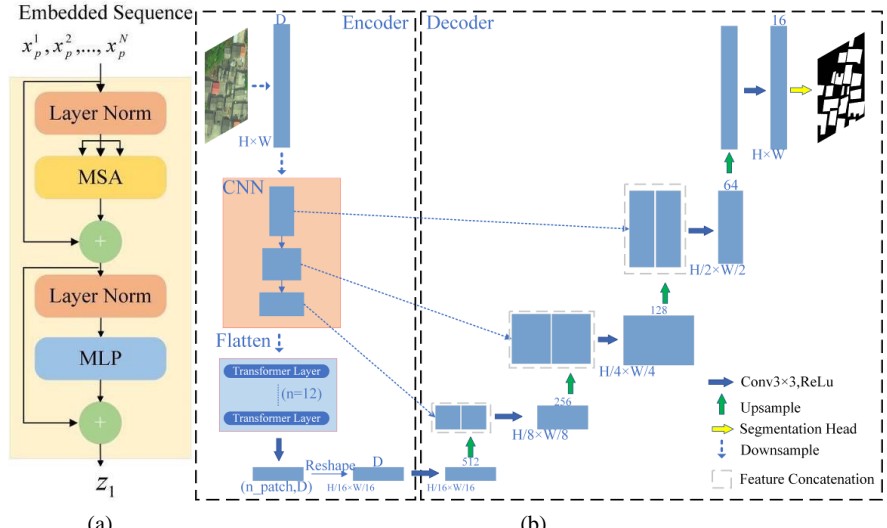

334 (a)  (b)

335 **Fig. 3.** The overview of TransUNet framework (adapted from (Chen et al., 2021)): (a) Schematic
336 diagram of Transformer layer; (b) Structure diagram of TransUNet


338 *3.3. Building's height acquisition*

339 UAV tilt photography modeling technology can combine control points encryption from
340 massive image data with a small number of ground control points to obtain accurate external
341 orientation elements (Kang et al., 2020). The conducted 3D model reflects the truly condition of
342 the ground, and the data is selected to be in the WGS 1984 coordinate system. The ground
343 resolution is one of the most intuitive and important parameters in tilt photography, and it's also a
344 key factor determining the quality of the 3D modeling. In the process of performing aerial
345 triangulation for tilt-image automation, it is necessary to ensure that the resolution of the different
346 images is as consistent as possible while taking into account the resolution of the side-view image,
347 thus to ensure accuracy and image overlap. Hence, the combinatory analysis of image resolution at
348 tilted viewing angle is required. The tilted image center point, near point and far point resolutions
349 are expressed as follows:

$$\text{GSD}_{\text{top}} = \frac{\delta h \cos \beta_y}{f \cos \left( \alpha_y - \beta_y \right)} \tag{3.12}$$

$$\text{GSD}_{\text{mid}} = \frac{\delta h}{f \cos \alpha_y} \tag{3.13}$$

$$\text{GSD}_{\text{bottom}} = \frac{\delta h \cos \beta_y}{f \cos \left( \alpha_y + \beta_y \right)} \tag{3.14}$$

350 Where $\delta$ is sensor cell size, $h$ is flight height, $f$ refers to the camera focal
351 length, $\alpha_y$ and $\beta_y$ are respectively dip angle and half the angle of view. Normally, the ground
352 resolution at the center of the tilted and vertical images should be comparable, and the minimum
353 resolution of tilted images should less than three times the resolution of a vertical image.
354 There are multiple formats available for storing 3D models, including OBJ, STL, FBX,



OSGB, etc. In this study, the generated 3D model is saved as OSGB format. OSGB format is
originally proposed by Ordnance Survey for storing the geographic spatial data in the British. It
combines binary encoding and compression algorithms to improve the data storage and
transmission efficiency. Normally, the OSGB data contains information of geographic coordinates,
elevation, texture mapping, and geometric shapes, which can be used to GIS application, virtual
reality (VR), among others.
Digital surface model (DSM) is a digital terrain model that contains more elevation
information about trees, buildings, and bridges. Compare to DEM, DSM can reflect the truly
surface condition of earth, thus DSM has a wide range of application in city management or forest
stewardship. In this research, the UAV data can be transformed to DSM data by using SuperMap
software, and the DSM result is shown in Fig. 4(b). After generating the DSM, the elevation of the
roof of the building and the corresponding elevation of the ground around the building are
extracted by manual selection, then the height of buildings can be calculated by using equation
(3.15).

$$DSM_{Roof} - DSM_{Ground} = H \qquad (3.15)$$

Where $DSM_{Roof}$ is the DSM value of the building's roof, $DSM_{Ground}$ represents the
corresponding DSM value of ground, and $H$ is the result of building's height.

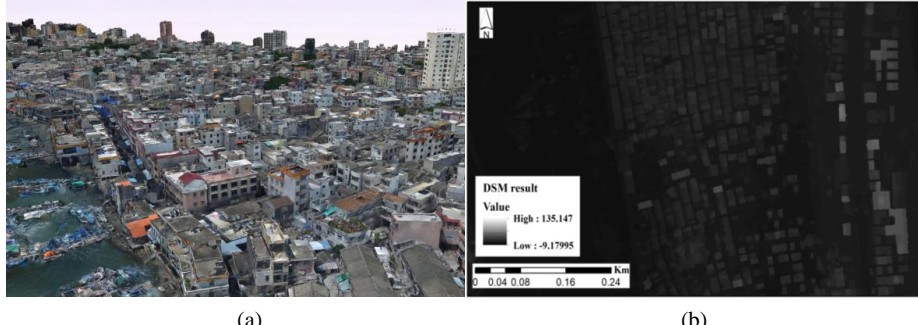

(a)                     (b)

**Fig. 4.** Building's height acquisition: (a) The schematic diagram of UAV tilt photography data; (b)
The generated DSM results for Building height data extraction.

*3.4. Exposure and vulnerability assessment*
The process of storm surge risk assessment involves two key components: exposure and
vulnerability. The exposure represents the elements exposed to hazardous spaces, while the
vulnerability refers to the level of the exposure elements' susceptibility to damage. When doing
exposure assessment, the disaster-affected elements can be conducted by overlaying the building
footprint data and land cover data with the hazard layer, which is the inundation data in this
research. The process can be accomplished using overlay analysis in ArcGIS software.
*3.4.1. Adaptation of flood vulnerability functions.*
Constructing an empirical stage-damage curve is a commonly used method for conducting
vulnerability assessments. However, as is mentioned above, China lacks of the data about flood
loss or insurance compensation in flood disasters, as a result, it's not practicable to develop
exclusive functions for the study region, so the depth-damage functions developed by Huizinga,
Joint Research Center (JRC) (Huizinga et al., 2017) are introduced. The depth-damage functions



manifest the loss ratio of the exposure elements in different inundation depth from 0 to 6 m, and
the ratio range from 0 to 1, which represents no damage to fully damaged. Besides, JRC also
provides the maximum economic losses per square meter for six different exposure element types
including residential, industrial, infrastructure, road, agricultural land, and transport. In this study,
the original functions and maximum loss data for China region are used, and the economic loss
can be calculated by multiplying the loss ratio, the maximum loss, and the disaster-affected area.

The building's height is an important factor in flood loss estimation, normally the damage
ratio decreases as the number of floors increases (Taramelli et al., 2022). However, the JRC's
vulnerability functions do not provide the specific function of each height category. In this case,
the depth-damage functions in HAZUS are introduced. HAZUS is first released for earthquakes in
1997 by Federal Emergency Management Agency (FEMA), and that's when the HAZUS Flood
Model started to be developed (Scawthorn et al., 2006). In 2004, a multi-hazard version called
HAZUS-MH was a standardized GIS-based model that included the earthquake, flood, and
hurricane models (Nastev and Todorov, 2013). The HAZUS-MH flood model is designed
primarily for local and regional hazard planners and emergency managers for developing
emergency management plans and mitigation strategies (Tate et al., 2015). However, the
depth-damage functions in HAZUS-MH are restricted to regions within America, hence the
HAZUS's functions are introduced to adapt JRC's functions.

The approach to modifying functions is refered to the method proposed by Dabbeek et al.
(2020). In the process, the HAZUS loss ratios of each height category (one-story, two-story, three
and more-story) are averaged, which is shown in equation (3.16). Then the contribution of each
height category relative to the average loss is calculated as equation (3.17) shows. In the end,
multiplying the value obtained in the previous step by JRC's vulnerability functions yields the
adapted functions for each height category.

$$\bar{D}_{i(hazus)} = \frac{d_{i,1} + d_{i,2} + d_{i,3+}}{n} \quad i(depth) = \{(0,6)\} \tag{3.16}$$

$$c_{i,h} = \frac{d_{i,h}}{\bar{D}_{i(hazus)}} \tag{3.17}$$

$$d_{i,h(adapted)} = c_{i,h} \times \bar{D}_{i(jrc)} \tag{3.18}$$

Where $d_{i,h}$ represents to the loss ratio at the inundation depth $i$ for each height category
$h$. $\bar{D}_i$ is the average loss ratio of all heights.
*3.4.2. Quantitative risk assessment*
The quantitative financial loss estimation is accomplished by overlaying the following data:
the inundation simulation result generated by FVCOM and SWAN modeling, the spatial
distribution of three types of exposure elements, the depth-damage functions of industrial and
commercial elements, and the adapted depth-damage functions for residential elements in three
height categories. The process of loss estimation can be shown in the following equation:

$$C = \sum_{i=1}^{i=n} D_{x(i)} f(d_i) A_i \tag{3.19}$$

Where $C$ stands for the economic loss estimation result. $n$ represents the total number of
exposure elements. $x(i)$ is the type of the i-th element and $D_{x(i)}$ is the maximum loss of the i-th
element. $d_i$ is the depth of submergence of the i-th element and $f(d_i)$ is the loss ratio of the i-th




element. $A_i$ refers to the area of the i-th element.

Comparing to the 984 euros per m² monetary loss of residential buildings in 2010, the

monetary loss of infrastructure and agriculture are respectively 12 euros per m² and 0.02 euro per
m² according to JRC, only account for 1% or less. Therefore, the monetary loss estimate of
infrastructure and agriculture is excluded in the study.

In this research, five storm surge scenarios are settled, ten administrative sub-zones are given

four different risk levels for each defined typhoon scenario.

**4. Results and discussions**
*4.1. Validation*

The performance of coupled FVCOM-SWAN model is evaluated. Four typical typhoons

(Vicente, Hato, Mangkhut, Khanun) are selected to validate the coupled model for the study region.
The measured data of each typhoon are captured by Department of Natural Resources of Huizhou
Bureau. Fig. 5 shows the maximum predicted water level and highest measured water level of the
chosen typhoons. Relative error and absolute error are introduced to evaluate the model and Table
1 displays the statistical results from Huizhou tidal station. It is seen that the predicted results are
in good agreement with the measurements. The statistic result shows that the relative errors of the
four typhoons range from 2.1% to 19.8%, and the absolute error varies from 4 cm to 54 cm.
Therefore, the coupled FVCOM-SWAN model demonstrates a reliable competence in
accomplishing the storm surge simulation task.

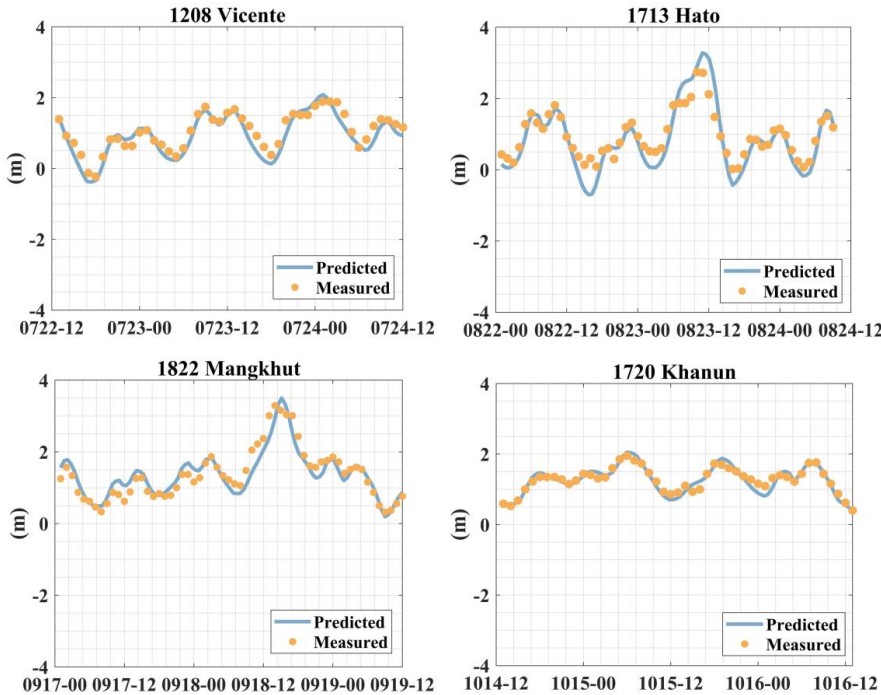

**Fig. 5.** The predicted water level and highest measured water level recorded by Huizhou tidal station during different typhoon event




**Table 1.** The Relative error and Absolute error between maximum predicted water levels and highest measured water levels from Huizhou tidal station during different typhoon events

| Typhoon name | Measured data (cm) | Relative error (%) | Absolute error (cm) |
| --- | --- | --- | --- |
| Vicente (1208) | 189 | 10.3 | 19 |
| Hato (1713) | 274 | 19.8 | 54 |
| Mangkhut (1822) | 329 | 6.5 | 22 |
| Khanun (1720) | 201 | 2.1 | 4 |

*4.1. Hazard assessment*

In the present research, five storm surge inundation scenarios are defined, which represent five different typhoon return periods: 10-year, 20-year, 50-year, 100-year, 1000-year respectively corresponding to minimum central pressure 940hPa, 930hPa, 920hPa, 910hPa, 880hPa, and the probability of occurrence are 10%, 5%, 2%, 1%, 0.1%. The simulation result is displayed through ArcGIS 10.8 software, and the inundation area and depth simulation results for each scenario is shown in Fig. 6. It is seen that the inundation area is spread over the coastal area in southwest of study area. In particular, for the 1000-year return period scenario, the inundation area exceeds 13 km² in the study area. Moreover, the presence of Double-Moon Bay leads to the extension of the inundation along the bay, contributing to severe disasters inland.

From the point of view of different scenarios, the area of inundation in direct proportion to the typhoon's return period, and the proportion of inundation area increases from 14% to 31% of study area. When the return period is less than 50 years, most of the flooded area is considered to be in a high-level hazard zone, accounting for 75% for a 10-year return period and 67% for a 20-year return period, and no zone in very high-level hazard. Basically, the inundation area covers land such as grassland, saline land, and some buildings near the estuary as the area is more susceptible to flooding because of the lower elevation and drainage from the estuary. As the return period goes up to 100 years, 34% and 36% of the flooded area are defined at a high-level hazard and very high-level hazard. When it's 1000-year, the situation worsens with approximately half of the inundation area being considered in very high-level hazard. Typically, the flood extends from the margin of terrene, however, the southernmost region of the investigated area is characterized by a knoll covered by forest vegetation, which serves the dual purpose of water absorption and flood mitigation. In addition, the construction of embankments on both sides of Double-Moon Bay effectively withstands flooding. Nevertheless, because of the presence of the estuary, inadequate water absorption ability of coastal saline soil and the hydrological system, the inundation flows in through the estuary and spreads inland.
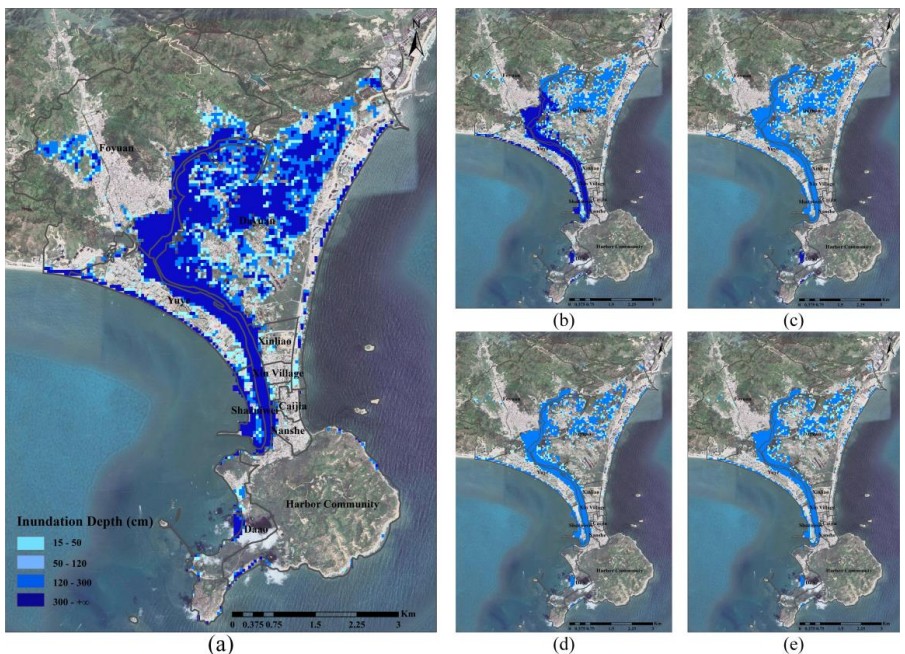

**Fig. 6.** The storm surge inundation simulation results of different typhoon scenarios. The base map is obtained from © GoogleMaps (map data © 2023 Google).

*4.2. Buildings' characters extraction*

Buildings are places where human populations gather and distribute, and contain amounts of property, which have great significance in quantitative risk assessment.

*4.2.1 TransUNet model training*

The dataset construction area is chosen at southwest waterfront region of Renshan Town. The specific location is shown in Fig. 7. The chosen area is a typical area of the Huizhou coastal area. Apart from the seaside bungalows, the area contains some high-rise buildings that are identified as commercial hotels or resorts, while dense residential area is also widely distributed throughout the inland region. In conclusion, the chosen area contains different kinds of buildings with strong representativeness. Since most of the buildings in China coastal towns have the similar characters, the model trained on the representative region has the ability to identify buildings in other regions rapidly.

The labels of the buildings in the area are generated by manually annotation, and the image is cropped to pixels with a size of 256*256. Besides, some of the images without buildings are filtered for preventing the effect of imbalance between the building samples and background samples. In the end, a total of 1200 labeled building dataset is constructed, and the dataset size is deemed sufficient when compared to previous study (Dixit et al., 2021; Ji et al., 2018). The dataset is then divided into a training set and a test set, with the ratio of 8:2. Data enhancement techniques, such as random hue saturation value, random shift scale rotate, flip, and rotate, are implemented during model training to improve the deep learning model's generalization performance and prevent overfitting.


The training's initial learning rate is set to 1e-5, and the learning rate adjustment strategy for
improved training. The batch size is specified as 4, and the number of training epoch is 100. The
model is trained on a NVIDIA RTX3060 GPUs.

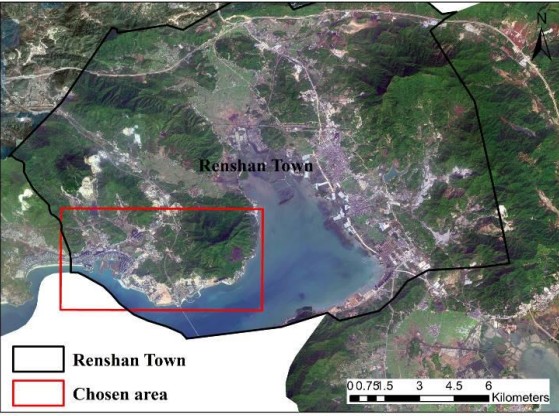

**Fig. 7.** The chosen area to make the training samples. The base map is obtained from ©
GoogleMaps (map data © 2023 Google).

*4.2.2. Extraction result*
Several effective indicators are introduced, including Recall, Precision, F1-score, and mean
Intersection-over-Union (mIoU), to evaluate the performance of the deep learning model. Recall is
the probability of being predicted as positive among actual positive samples. Precision, on the
other hand, is the probability of being actually positive among samples predicted as positive.
F1-score serves as an indicator that achieves a balance point between precision and recall,
essentially being the harmonic average of precision and recall. mIoU is the mean ratio of the
intersection to the union between predicted and true values for each category. True positive (TP)
indicates the true samples that are predicted correctly by the model. False positive (FP) indicates
the positive samples that the model incorrectly predicted. True negative (TN) and false negative
(FN) refer to the number of samples that are correctly and incorrectly predicted as negative by the
model. The equations of Recall, Precision, F1-score, and mIoU are as follows:

$$recall = \frac{TP}{TP + FN} \tag{4.1}$$

$$precision = \frac{TP}{TP + FP} \tag{4.2}$$

$$F1 = 2 \times \frac{precision \times recall}{precision + recall} \tag{4.3}$$

$$mIoU = \frac{1}{k+1} \sum_{i=0}^{k} \frac{TP}{TP + FP + FN} \tag{4.4}$$

The quantitative evaluation result is shown in Table 2, and the visualization results are
illustrated in Fig. 8. As Table 2 shows, the recall score reaches 87% indicating that most of the true
building pixels are predicted correctly, and Precision indicates that 82% of all building pixels are
correctly detected. Moreover, both the mIoU score and F1-score exceed 80% manifest that the
model can balance well between precision and recall. These results reflect the strong performance





of TransUNet in the building extraction task. After post-processing the result, such as boundary
simplification, the building vectorization results can be used for further research in risk assessment.
The overall result is shown in Fig. 9.
**Table 2.** The statistical accuracy assessment of building footprint extraction

| Evaluation metric | |
|---|---|
| Recall (%) | 87.03 |
| Precision (%) | 82.04 |
| F1-score (%) | 84.46 |
| mIoU (%) | 83.38 |

|        |        |        |
|--------|--------|--------|
| (a)    | (b)    | (c)    |


**Fig. 8.** Building footprint extraction result in study area. (a) Remote sensing images obtained from
Jilin-1 satellite (© Chang Guang Satellite technology CO., LTD); (b) Extraction result; (c) Ground
truth. The building is marked in white, and the background is marked in black

*4.2.3. Building height calculation*
Through combing two methods mentioned above, the height information is acquired in units
of meters. The number of floors is derived by dividing the acquired height information by the
specified standard height of 3 meters, according to the China residential design standards. The
general condition of building floor is shown in Table 3. Just as mentioned above, the buildings in
study area are mainly for residential and commercial use. Since the study area is undeveloped, the
high buildings and large mansions is relatively less common, and most of them are built for
seaside resort. Instead, buildings with 5 floors or less are the mainstream in study area as respected,
which the proportion can reach 76.5%. The building footprint extraction result and building's
height information extraction result can be found in Fig. 9.
**Table 3.** Statistical results of building height in the study area

| Building floor | Area (m²) | Proportion (%) |
|---|---|---|
| 1-5 | 17537238.61 | 76.5 |





| | | |
|---|---|---|
| 6-10 | 4996897.08 | 21.8 |
| 11-20 | 342207.82 | 1.5 |
| 20+ | 54083.93 | 0.2 |

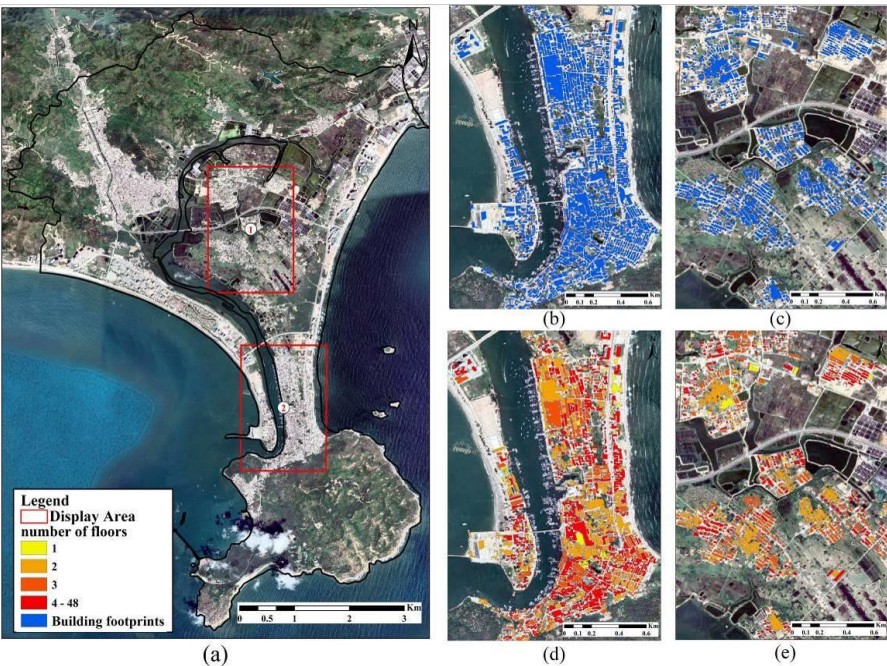

**Fig. 9.** The building characters extraction result: (a) The schematic of the display area; (b, c) Building footprint result in area 1 and 2; (d, e) Building height result in area 1 and 2. The base map is obtained from © GoogleMaps (map data © 2023 Google).

*4.3. qualitative risk assessment*

Risk matrix is a risk assessment approach firstly developed by Electronic System Center, which was originally to assess the risk in the life cycle of purchase project (Garvey and Lansdowne, 1998). An additional qualitative risk assessment is conducted using the risk matrix method, incorporating improved land use data to highlight the superiority of building extraction in flood risk assessment.

As is shown in Fig. 10(a), the concentrations of organic town of Dayuan village and Shazuiwei makes it in very high vulnerability level. Under the circumstance of defined 880hPa storm surge scenario, the inundation area spread inland which makes the majority area of Dayuan is regarded as moderate risk, and a fraction of the only very high risk area is distributed in Shazuiwei and north of Dayuan village. In the area of Yuye village, part of the south coastal area is considered in moderate or high risk level. That is mainly because the majority area of Yuye is defined as resort district except for a few areas of tidal flats, which is in high vulnerability. However, after referring to the result of hazard assessment, buildings in the area are not actually inundated, meaning the area should not be at risk level.

Through comparing the Fig. 10(a) and Fig. 10(b), the enhanced land use data in the present




research demonstrates a higher ability to recognize vulnerability elements, which the type is
buildings in the present research. The two red boxes in the figure highlight the noticeable disparity
between the original and current results. The present risk assessment provides more refined risk
assessment result compared to the original result, as the previously identified large hazardous
areas are replaced with more detailed and smaller zones. This refinement is conducive for
government or decision-makers to conduct disaster prevention measures, propose quick guidance
for personnel evacuation and organize rescue operations in the event of a disaster.

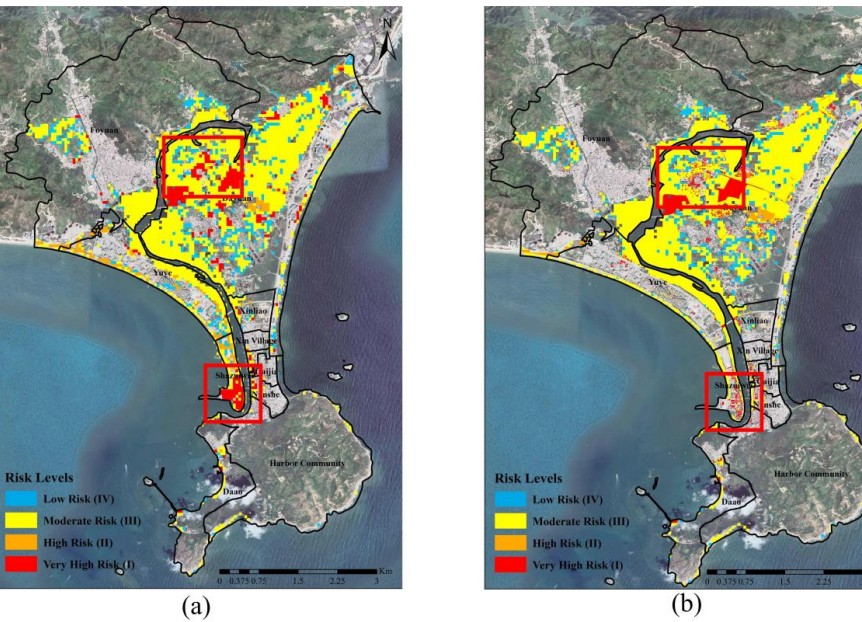

(a)                                                    (b)


**Fig. 10.** The risk assessment maps before (a) and after (b) improvement for storm surge scenarios
of 1000-year return period. The base map is obtained from © GoogleMaps (map data © 2023
Google).

*4.4. JRC's depth-damage function adaption*
Fig. 11 illustrates the damage ratio given flood-depth after adjustment, respectively for one-,
two- and more than three-story residential buildings. After adjustment, the damage of one-story
residential building function is significantly enhanced, and the loss ratio reach 1 early, which is
explicable as 2m-depth flood almost submerges the entire building, resulting in a potential loss of
the maximum property value. On the contrary, the loss ratio for multi-story residential building is
decreased relative to the original function, it reaches the same level as in the original function
when the water depth reaches 5 meters. Furthermore, the function of a two-story residential
building is quite similar to that of a building with three or more stories. This can be attributed to
the flood's effect on buildings with six meters or less depth being nearly the same, on account of
the flood can't overwhelm the entire buildings.
Although JRC provides the maximum monetary damages, they are computed for Beijing in
2010. However, there is a substantial difference in the level of development between Beijing and
the study area. For better matching the financial level in study area, adjustment can be achieved

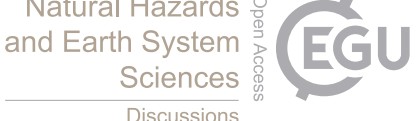



based on scaling the maximum monetary damage value with the GDP ratio according to Huizinga
(2007). Based on the 2010 GDP of Beijing of 14113558 million yuan and the GDP of Huizhou of
172995 million yuan, the maximum monetary damage is adjusted by equal proportions. Besides,
the price level also needs to be adjusted to the 2022 price level. According to the World Bank, the
Chinese consumer price index (CPI) has changed from 100 in 2010 to 131.9 in 2022, the tendency
of variation and the adjusted maximum monetary damages are shown in Fig. 12.

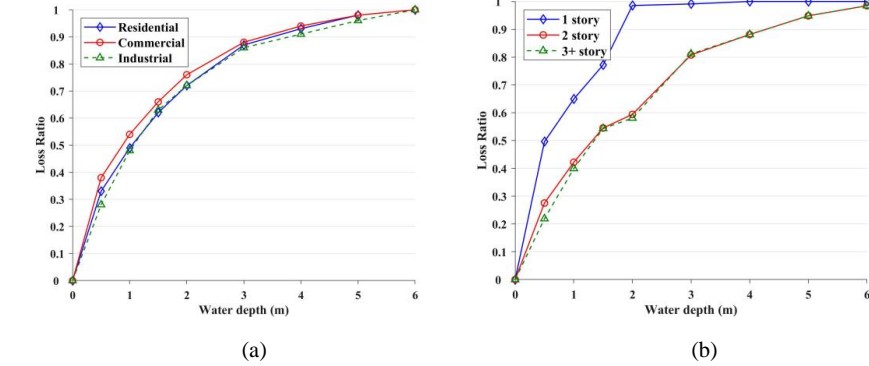

(a) (b)
**Fig. 11.** (a) The depth-damage function proposed by JRC; (b) The adapted depth-damage function
for residential buildings in different floors

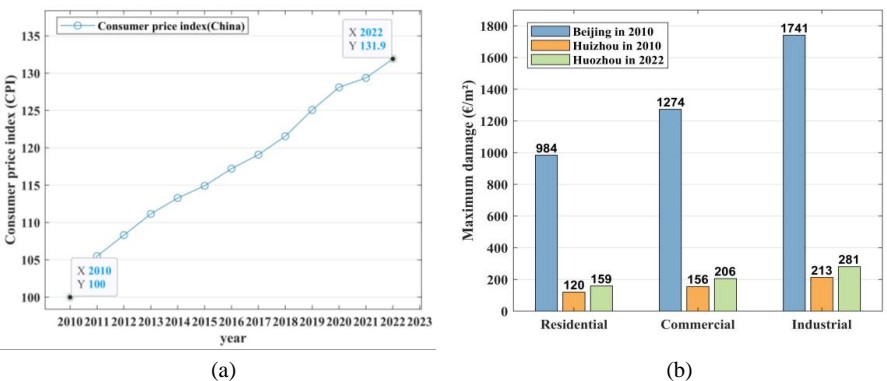

(a) (b)
**Fig. 12.** (a) The variation trend of Consumer price index released by World Bank; (b) The
maximum monetary damage per m$^2$ for each type of exposed elements in China (in 2010 and in
2022).


*4.5. Quantitative risk assessment*
Loss assessments of five storm surge scenarios are computed for return periods of 10, 20, 50,
100, and 1000 years, through employing the method in section 3. The estimate monetary damage
is summarized in Table 4.
The statistical data in Table 4 demonstrate an increase in the affected area and total economic
loss with an increasing return period. Comparing to the total affected area of 131533.12 m$^2$ and
the total economic losses of 9330517.49 euros with the 10-year return period, the

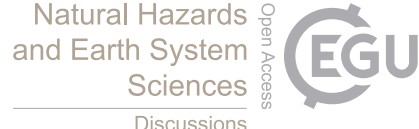

corresponding estimate result with 1000-year return period is 917437.99 m$^2$ and 68364923.25
euros, which is both approximately seven times higher. This indicates a proportional
relationship between the extent of regional impairment and the return period of a typhoon.
Although the impacted area for the 20-year and 50-year return periods exhibits relative
proximity as the different is 24118.26 m$^2$, there is still a significant disparity in economic
losses. According to the inundation result above, that's because the inundation area of two
return period is nearly the same except for the slight difference in the northeast of the study
region, but the flood depth of 50-year intensified, which causes more monetary damage. In
terms of inundated building types, in case that study area is characterized as a tourism and
fish breeding area, the proportion of economic losses in industrial is relatively low. The losses
of residential buildings and commercial buildings is comparatively close, up until the severity
of storm surge reach 50-year return period. At this point, the losses experienced by residential
buildings exceed those incurred by commercial buildings by more than double. The fact can
be explained by the commercial buildings area mainly constructed by the seaside for better
turnover therefore both type of waterfront buildings is impacted. However, as the severity of
the typhoon worsens, more residential settlements inland are flooded, resulting in a swift
increase in economic losses for residential buildings.
**Table 4.** The statistic result of the quantitative risk assessment for five defined typhoon scenarios.

| Scenario | Elements | Area (m$^2$) | Economic losses (€) | Total losses (€) | Probability | Risk (€) |
|---|---|---|---|---|---|---|
| 10-year (940hPa) | Residential | 94847.11 | 4910882.27 | 9330517.49 | 0.1 | 933051.75 |
| | Commercial | 36163.62 | 4281840.09 | | | |
| | Industrial | 522.39 | 137795.12 | | | |
| 20-year (930hPa) | Residential | 216010.31 | 7872861.19 | 13665211.91 | 0.05 | 683260.60 |
| | Commercial | 55423.59 | 5602828.01 | | | |
| | Industrial | 522.39 | 189522.71 | | | |
| 50-year (920hPa) | Residential | 237572.35 | 16509796.15 | 24607011.73 | 0.02 | 492140.23 |
| | Commercial | 57979.81 | 7775321.70 | | | |
| | Industrial | 522.39 | 321893.88 | | | |
| 100-year (910hPa) | Residential | 291759.48 | 19857901.69 | 28446797.47 | 0.01 | 284467.97 |
| | Commercial | 75123.51 | 8194736.70 | | | |
| | Industrial | 833.39 | 394159.08 | | | |
| 1000-year (880hPa) | Residential | 762570.09 | 49295364.67 | 68364923.25 | 0.001 | 68364.92 |
| | Commercial | 149457.01 | 17907591.59 | | | |
| | Industrial | 5410.89 | 1161967.00 | | | |


Based on the economic losses estimation result for five storm surge scenarios, through using
zonal statistics method on the data of administrative sub-zones in the study area, the quantitative
risk assessment is conducted. The economic losses and spatial distribution of storm surge risk for
ten sub-zones in five different scenarios are shown in Fig. 13. The zonation statistics result map of
each sub-zone is defined at four different risk levels (very high, high, moderate, low). The
classification of risk levels is obtained by categorizing all zonal statistic result based on quantiles.
As is shown in Fig. 13, Dayuan village is considered in very high risk for every defined
typhoon scenario. Through analyzing the geographical characteristics of the study area, it can be



found that although Dayuan is a relatively inland village, it's surrounded by the watercourse of the
estuary of Double-Moon Bay. Due to the existence of flood control dam, both side of the bay offer
a measure of protective effectiveness, which result in the water level rises in inland watercourse,
and further causes flooding of residential buildings in Dayuan village, leading to massive financial
losses. In contrast, Foyuan village is also a village with a relatively large area, the risk is at
moderate level for 10, 20-year return period, and the level escalates to high for 50, 100-year return
period and reaches very high in 1000-year. Considering the presence of the knoll, the spread of
inundation is hindered. However, as typhoon becomes more severe, the inundation hit the western
buildings in the region, which led to the phenomenon of progressively escalating risk level. In
terms of those villages with relatively smaller sizes, due to the protection of dam, Xinliao village,
Xin village, Caijia village all are defined in relatively low risk level, although the regions with a
high density of buildings. Shazuiwei and Yuye village in different return period are considered in
different risk level, the cause of this phenomenon might be that apart from the higher density of
buildings, the buildings in Shazuiwei are distributed in coastal area, combing the impact of
inundation of both sides as it's located at the outermost part of the gulf. Consequently, the risk
level in Shazuiwei remains consistently high as opposed to gradually increasing like in Yuye
village. Although they are located at the outermost part of the study area, the quantitative risk level
of Daao village and Harbor community is gradually increasing for different return period, but it's
not as serious as the other village, which can be explained that these locations exhibit elevated
topography.

The quantitative risk assessment and zonal risk maps can assist the government or decision-
makers in recognizing the specific economic losses of each sub-zones, so it's helpful to identify
the areas that are more susceptible to experiencing significant losses, which allows them to
develop disaster prevention measures, for example constructing disaster prevention facilities,
budget allocation for disaster prevention and planning evacuation strategies. Besides, establishing
the quantitative risk for different typhoon periods can enhance the decision makers understanding
of the potential vulnerability in each sub-zone, and facilitates the implementation of appropriate
preventive and disaster relief measures facing different typhoon intensity.
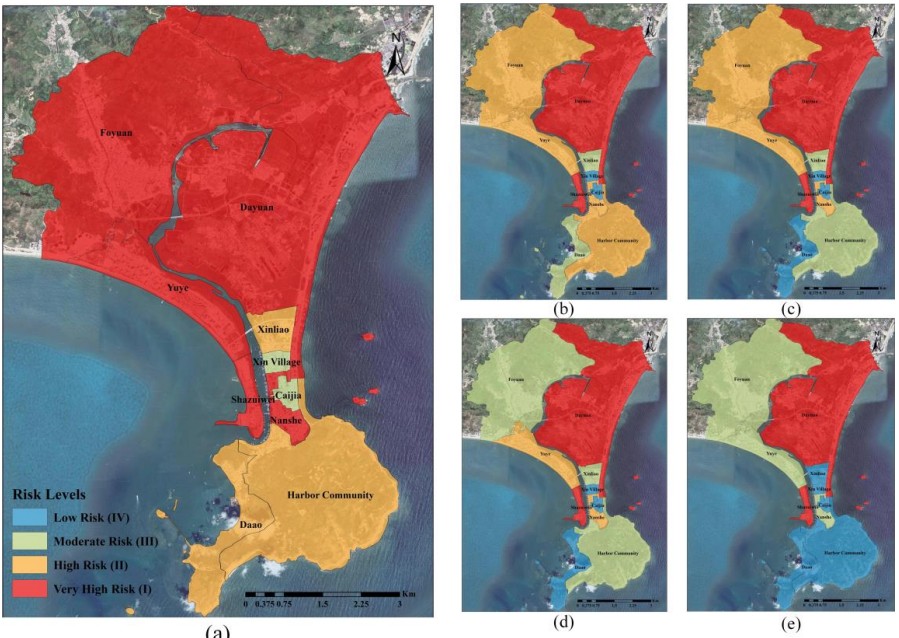


**Fig. 13.** The zonation maps of the quantitative risk assessment for five defined typhoon scenarios:
return period (a) 1000-year, (b): 100-year, (c): 50-year, (d): 20-year, (e): 10-year. The base map is
obtained from © GoogleMaps (map data © 2023 Google).

**5. Conclusions**

These years, the academic research on storm surge risk assessment has been greatly
developed due to climate change and financial growth in coastal area. However, the quantitative
risk assessment is inexecutable in the undeveloped area since on account of the lacks of building
characters and damage assessment data. Target at the question above, the purpose of this paper is
to propose a method for conducting refined storm surge risk assessment quantitatively based on
deep learning and GIS techniques. Firstly, the reliable coupled FVCOM-SWAN model is utilized
to simulate five defined storm surge scenarios. Facing the challenge of the absence of data, a deep
learning method TransUNet is applied to extract the building footprint data for refined extraction
of exposed elements, and buildings' height data is acquired through UAV. To compensate for that
the available depth-damage functions do not taking building's height into account, the functions
are adjusted for buildings with different floor and consequently to perform more refined monetary
losses calculations in five defined scenarios. Eventually, the quantitative risk assessment and
zonation maps of the study area are generated base on GIS techniques

The quantitative risk assessment result of the study region shows that on account of the
existence of estuary and the gathering of buildings, Dayuan village presents the high-risk level in
all defined typhoon scenario, and the economic loss risk is large. The flood control dam provides
protection of Xinliao village, Xin village, Caijia village, which prevents the regions suffering large
economic losses as the typhoon return period is 10-year and 20-year. However, the storm surges,
under the typhoon scenarios that the return period is greater than 50-year, can overwhelm the
existed dikes, and both the commercial buildings and residential buildings suffer heavy economic



losses. Therefore, it's necessary to make land use planning and adjustment especially in Dayuan and Shazuiwei as they are under very high-risk level to prevent the impact and losses caused by storm surges. Besides, the regions that is nearest to the sea doesn't mean they suffer greater potential economic loss, as the risk level of Daao village and Harbor community are considered at a relatively low level because of the topographical characteristics and the distribution of buildings.

The study provides a framework for refined quantitative storm surge risk assessment targeting the problem of acquiring exposure elements and the establishing multi-variable empirical depth-damage functions, as a consequence of missing data in underdeveloped regions. The generated results can help the decision-makers to identify the areas that are susceptible to experiencing significant losses efficiently, and help the respective authorities with disaster prevention, future land use planning and material deployment. Furthermore, it is important to remark that, the methodology of this paper has general applicability, since the applied models are publicly available. Thus, there is also potential for further application. For example, the framework can be applied in other coastal areas in China, as they have similar characters, which also means there is a possibility to utilize in larger scales. Furthermore, the framework can also be performed in other types of disasters, such as flood, earthquake, and mudslide. Consequently, the proposed methodology demonstrates an extensive relevance to the scientific community.

There is still room for improvement in this study. The current study relied on manual labeling in terms of distinguishing between functional areas to conduct risk assessment. In the future study, efforts will be made to distinguish the types of exposure elements in a more objective way, based on diverse data sources such as social media Point Of Interest (POI). Additionally, exploring the activity patterns of the population through multiple sources of data including taxi trajectories and smart cards can contribute to the consideration of population risks in different storm surge scenarios, thereby prompting more comprehensive risk assessments.

**Data availability.**

Remote sensing images are obtained from Chang Guang Jilin-1 satellite. The dataset of wind field is generated by ERA5 and Holland method. The Administrative Boundary data is obtained from National geographic information public service platform. The datasets can be obtained from https://dx.doi.org/10.6084/m9.figshare.24586605 (Yu, 2023). DEM data and UAV data are obtained from Department of Natural Resources of Huizhou Bureau, data sets are not publicly available due to the policy of the Natural Resources of Huizhou Bureau.

Competing interests. The authors declare that they have no conflict of interest.

**Authorship contributions.**

**Lichen Yu**: Investigation, Methodology, Data Curation, Visualization, Formal analysis, Writing - Original Draft. **Shining Huang**: Investigation, Data Source, Data Curation, Visualization, Formal analysis. **Hao Qin**: Conceptualization, Methodology, Validation, Supervision, Writing - Original Draft, Project administration, Funding acquisition. **Wei Wei**: Validation, Data Curation, Visualization. **Lin Mu**: Conceptualization, Supervision, Project administration, Funding acquisition.

**Acknowledgement**





This work was supported by the Shenzhen Science and Technology Program (Grant No. KCXFZ20211020164015024), National Key Research and Development Program of China (Grant No. 2021YFC3101800), National Natural Science Foundation of China (Grant No. 52101332 and U2006210).

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
