# Peer review of "Quantitative Study of Storm Surge Risk Assessment in Undeveloped Coastal Area of China Based on Deep Learning and Geographic Information System Techniques: A Case Study of Double-Moon Bay Zone"

_Natural Hazards and Earth System Sciences, 2023_

## Author Comment (AC1)

Dear Editor and Reviewer:

Thank you very much for your supervision of the reviewing process of our manuscript. We highly appreciate your carefulness and broad knowledge on the relevant research fields. The article has been revised according to your constructive comments and valuable suggestions. The main responses to the comments are described as follows:

**1.** The paper utilizes FVCOM-SWAN model to complete the simulation of the storm surge inundation by inputting the modified parameters of Typhoon Mangkhut, and further to generate the risk map. How to explain the generated result can help manage the future storm surge? Regarding the selection of the five typhoon scenarios, there isn't a clear statement about how did the authors decide on those return periods. Therefore. I would recommend to add more information about the setup of the scenarios.

**[Reply]**

The potential storm surge inundation maps in different typhoon scenarios have been conducted by institutions such as the National Oceanic and Atmospheric Administration (NOAA), National Hurricane Center, and other departments since the 1990s (Glahn et al., 2009). In the field of risk assessment research, it is common to set up different typhoon scenarios using storm surge simulation models to obtain various scenarios of typhoon induced inundation (Zhang et al., 2023; Rizzi et al., 2017). The hazard maps of under various typhoon intensity scenarios are helpful for decision-makers and researchers in analysing multiple aspects of potential hazards in the study area.

Typhoon Mangkhut, as one of the largest typhoons to affect South China Sea region in recent years, has a strong representative. It is characterised by high intensity, wide area of influence, high wind speed, etc. In this study, the path of Typhoon Mangkhut is shifted to pass through the Huizhou tidal station as the input typhoon path of the coupled model to maximize the impact area of the simulation result. In terms of the center pressure, Wang et al. (2021) presented statistical analyses of historical typhoon data in Huizhou, and designed five typhoon scenarios, which are respectively the typhoon minimum central pressure of 880, 910, 920, 930 and 940 hPa. Therefore, these five parameters are introduced as the setup for five typhoon scenarios.

In the context of global warming and increased climate extremes, the occurrence of large-scale typhoons has become more frequent, such as Typhoon Rammasun and Typhoon Meranti (corresponding to 100-year return period). Therefore, the modified typhoon parameters are utilized for simulation five typhoon scenarios, in order to assuming the different storm surge disaster situation in the future.

The first two paragraphs have been added to section 3.1 of the revised manuscript, and the third paragraph has been added to the Conclusions section of the revised manuscript.

**2.** The article uses UAV data in the realm of buildings' height data acquisition. Whether the same purpose can be achieved by using publicly available DSM data, or methods base on satellites such as building shadow length calculation in remote sensing images? I would suggest to add some explanations about the advantages of utilizing this method.

**[Reply]**

When the building is inundated, there are a variety of factors that may influence the amount of

monetary loss. For example, building type, building structure, private precaution, maintenance status, and others (Marvi, 2020; Thieken et al., 2008). Taramelli et al. (2022) pointed out that building's height is one of the factors for determining the susceptibility due to flooding and evaluate the buildings' potential damage by flood hazards. Hasanzadeh Nafari et al. (2016) developed a new loss model, in which building with different story were divided into different categories in the modelling process. To conclude, height is an important factor that affecting the vulnerability of buildings when they serve as inundation-exposed elements. Therefore, in the process of quantitative storm surge risk assessment, it is necessary to adjust the depth-damage functions to make buildings of different heights correspond to different functions.

Besides, different from the field research and statistics required for other data acquisition, the data of buildings' height is more accessible from multiple sources. For example, public data DSM data has been utilized for building height estimation (Huang et al., 2022), some satellite companies also offer services to customize DSM data for selected regions. Nonetheless, they respectively suffer from a lack of precision and high costs. Building height can also be obtained via remote sensing technique, such as Synthetic Aperture Radar (SAR) (Li et al., 2020; Frantz et al., 2021), or take advantage of shadow in remote sensing images (Comber et al., 2011; Shao et al., 2011). However, in addition to the lack of precision, the absence of data necessary for modelling and the crowded character of rural buildings in China make the above methods difficult to be implemented. Compared to methods above, acquiring building height through UAV ensures high accuracy while being relatively efficient, and the method is quite simple, which also reduces the required costs.

These paragraphs have been added to the Introduction section of the revised manuscript.

**3**. In the result section, the authors conducted qualitative result for the new and old methods, but in the subsequent pages, the quantitative result is not correlated with the qualitative result. I would suggest adding more about the comparison of two results to the result section, thus better demonstrating the characteristics of quantitative assessment.

**[Reply]**

Comparing the qualitative risk assessment result and the quantitative risk assessment result, the first difference to be noticed is that the two results focus on different scales. For the qualitative result, the emphasis is on delineating the regions in different risk levels, which leads to the prevention and control of priority areas. Whereas for quantitative result, the scale of the result is limited to the village level zoning, as the estimated monetary loss amounts are summarized at the village level. Furthermore, while the qualitative results suggest that certain regions may not be at a moderate or high risk level, the quantitative result reveals that the estimated monetary loss for those villages are not insignificant. In conclude, the qualitative risk assessment provides new results from a completely different perspective than qualitative risk assessment. The results can provide intuitive information about the potential monetary loss to the secondary government departments, thus to help provide constructive suggestions in terms of risk prevention and control.

The paragraph has been added to section 4.5 of the revised manuscript.

**Specific modifications:**
**(1)** Line 182: "hotel", "resort" should be changed to plural form.

**[Reply]**

Thanks for your correction. This sentence has been revised based on your feedback.

**(2)** Line 185: the sentence is incomplete.

**[Reply]**

Thanks for your correction. This sentence has been revised to reflect that "However, the economic status of the region remains relatively low, which presents a challenge due to data scarcity and limited accessibility.".

**(3)** Line 218: the title should be "wind field data", and "Holland wind field" should be introduced concisely.

**[Reply]**

Thanks for your comment. The paragraph has been amended as follows:

(5) Hybrid wind field: ERA5 is the fifth generation of the European Reanalysis dataset produced by the European Centre for Medium-Range Weather Forecasts (ECMWF), and it provides the comprehensive and high-resolution atmospheric and climate data. Holland typhoon wind field model was proposed by Holland in 1980, which introduced Holland B parameter on the basis of the Schloemer exponential pressure distribution model (Holland, 1980). In this study, the two data are fused to generate hybrid wind field data, which is subsequently utilized for storm surge simulations.

**(4)** Line 228: The name of the data platform should be capitalized.

**[Reply]**

Thanks for your comment. This sentence has been revised to reflect that "the data is obtained from National Platform for Common Geospatial Information Services, and it contains administrative boundaries at village level.".

**(5)** Line 280: Modifications to the typhoon data should be presented in more detail.

**[Reply]**

Typhoon Mangkhut, as one of the largest typhoons to affect South China Sea region in recent years, has a strong representative. It is characterised by high intensity, wide area of influence, high wind speed, etc. In this study, the path of Typhoon Mangkhut is shifted to pass through the Huizhou tidal station as the input typhoon path of the coupled model to maximize the impact area of the simulation result. In terms of the center pressure, Wang et al. (2021) presented statistical analyses of historical typhoon data in Huizhou, and designed five typhoon scenarios, which are respectively the typhoon minimum central pressure of 880, 910, 920, 930 and 940 hPa. Therefore, these five parameters are introduced as the setup for five typhoon scenarios.

We have made revisions in the 3.1 section of the manuscript. We hope that this clarification effectively addresses your concern.

**(6)** Line 359: "elevation" should be changed to plural form.

**[Reply]**

Thanks for your correction. This sentence has been revised based on your feedback.

**(7)** Line 365-366: "elevation" should be changed to plural form.

**[Reply]**

Thanks for your correction. This sentence has been revised based on your feedback.

**(8)** Line 479: In the illustration of Figure 6 should contain the specific description of Figure 6(a-e).

**[Reply]**

Thanks for your comment. The caption of Figure 6 has been revised to reflect that "Fig. 6. The storm surge inundation simulation results of five different typhoon scenarios: return period (a) 1000-year, (b): 100-year, (c): 50-year, (d): 20-year, (e): 10-year. The base map is obtained from © GoogleMaps (map data © 2023 Google)."

**(9)** Line 483-484: Unnecessary statements, only results are discussed in the conclusion.

**[Reply]**

Thanks for your comment. The paragraph has been removed.

**(10)** Line 496: Incorrect expression of sentence. The image should not be cropped into pixels.

**[Reply]**

Thanks for your correction. This sentence has been revised to reflect that "The labels of the buildings in the area are generated by manually annotation, and the image is cropped into small patches with a size of 256*256.".

**(11)** Line 591: What is "the maximum monetary damages", a more specific explanation is needed.

**[Reply]**

Thanks for your comment. The Joint Research Centre provides information on the maximum damages per square for each type of building. This refers to the maximum monetary damage incurred when buildings are inundated, which is the monetary damage value when the damage ratio in the depth-damage curve reaches 100%.

We have made revisions in the section 4.4 of the manuscript. We hope that this clarification effectively addresses your concern.

**(12)** Line 602: "function" should be changed to plural form.

**[Reply]**

Thanks for your correction. This sentence has been revised based on your feedback.

**(13)** Figure 12: There is a redundant horizontal line in figure 12(a).

**[Reply]**

Thanks for your correction. The figure has been revised based on your feedback.

**(14)** Line 631: "type" should be changed to plural form.

**[Reply]**

Thanks for your correction. This sentence has been revised based on your feedback

**Reference:**

Comber, A., Umezaki, M., Zhou, R., Ding, Y., Li, Y., Fu, H., Jiang, H., and Tewkesbury, A.: Using shadows in high-resolution imagery to determine building height, Remote Sensing Letters, 3, 551-556, 10.1080/01431161.2011.635161, 2011.

Frantz, D., Schug, F., Okujeni, A., Navacchi, C., Wagner, W., van der Linden, S., and Hostert, P.: National-scale mapping of building height using Sentinel-1 and Sentinel-2 time series, Remote Sens Environ, 252, 112128, 10.1016/j.rse.2020.112128, 2021.

Glahn, B., Taylor, A., Kurkowski, N., and Shaffer, W. A. J. N. W. D.: The role of the SLOSH model in National Weather Service storm surge forecasting, 33, 3-14, 2009.

Hasanzadeh Nafari, R., Ngo, T., and Lehman, W.: Calibration and validation of FLFA$_{rs}$ -- a new flood loss function for Australian residential structures, Natural Hazards and Earth System Sciences, 16, 15-27, 10.5194/nhess-16-15-2016, 2016.

Holland, G. J.: An analytic model of the wind and pressure profiles in hurricanes, 1980.

Huang, H., Chen, P., Xu, X., Liu, C., Wang, J., Liu, C., Clinton, N., and Gong, P.: Estimating building height in China from ALOS AW3D30, ISPRS Journal of Photogrammetry and Remote Sensing, 185, 146-157, 10.1016/j.isprsjprs.2022.01.022, 2022.

Li, X., Zhou, Y., Gong, P., Seto, K. C., and Clinton, N.: Developing a method to estimate building height from Sentinel-1 data, Remote Sensing of Environment, 240, 10.1016/j.rse.2020.111705, 2020.

Marvi, M. T.: A review of flood damage analysis for a building structure and contents, Natural Hazards, 102, 967-995, 10.1007/s11069-020-03941-w, 2020.

Rizzi, J., Torresan, S., Zabeo, A., Critto, A., Tosoni, A., Tomasin, A., and Marcomini, A.: Assessing storm surge risk under future sea-level rise scenarios: a case study in the North Adriatic coast, Journal of Coastal Conservation, 21, 453-471, 10.1007/s11852-017-0517-5, 2017.

Shao, Y., Taff, G. N., and Walsh, S. J.: Shadow detection and building-height estimation using IKONOS data, International Journal of Remote Sensing, 32, 6929-6944, 10.1080/01431161.2010.517226, 2011.

Taramelli, A., Righini, M., Valentini, E., Alfieri, L., Gatti, I., and Gabellani, S.: Building-scale flood loss estimation through vulnerability pattern characterization: application to an urban flood in Milan, Italy, Natural Hazards and Earth System Sciences, 22, 3543-3569, 10.5194/nhess-22-3543-2022, 2022.

Thieken, A. H., Olschewski, A., Kreibich, H., Kobsch, S., and Merz, B.: Development and evaluation of FLEMOps – a new Flood Loss Estimation MOdel for the private sector, Flood Recovery, Innovation and Response I, 10.2495/friar080301, 2008.

Wang, S., Mu, L., Yao, Z., Gao, J., Zhao, E., and Wang, L.: Assessing and zoning of typhoon storm surge risk with a geographic information system (GIS) technique: a case study of the coastal area of Huizhou, Natural Hazards and Earth System Sciences, 21, 439-462, 10.5194/nhess-21-439-2021, 2021.

Zhang, S., Zhang, J., Li, X., Du, X., Zhao, T., Hou, Q., and Jin, X.: Quantitative risk assessment of typhoon storm surge for multi-risk sources, J Environ Manage, 327, 116860, 10.1016/j.jenvman.2022.116860, 2023.

---

## Author Comment (AC2)

Dear Editor and Reviewer:

Thank you very much for your supervision of the reviewing process of our manuscript. We highly appreciate your carefulness and broad knowledge on the relevant research fields. The article has been revised according to your constructive comments and valuable suggestions. The main responses to the comments are described as follows:

**1.** The use of building height to adjust the existing depth-damage functions is good. However, there is only few sentences in Line 394-395 about why building height is an important factor in quantitative storm surge risk assessment. Throughout the introduction, there is some description about the application of building footprint, but no introduction to the influence of building height. Thus, I suggest to introduce building footprint in more concise language, and meantime concurrently add some description of why building height was chosen as an important factor.

**[Reply]**

When the building is inundated, there are a variety of factors that may influence the amount of monetary loss. For example, building type, building structure, private precaution, maintenance status, and others (Marvi, 2020; Thieken et al., 2008). Taramelli et al. (2022) pointed out that building's height is one of the factors for determining the susceptibility due to flooding and evaluate the buildings' potential damage by flood hazards. Hasanzadeh Nafari et al. (2016) developed a new loss model, in which building with different story were divided into different categories in the modelling process. To conclude, height is an important factor that affecting the vulnerability of buildings when they serve as inundation-exposed elements. Therefore, in the process of quantitative storm surge risk assessment, it is necessary to adjust the depth-damage functions to make buildings of different heights correspond to different functions.

Besides, different from the field research and statistics required for other data acquisition, the data of buildings' height is more accessible from multiple sources. For example, public data DSM data has been utilized for building height estimation (Huang et al., 2022), some satellite companies also offer services to customize DSM data for selected regions. Nonetheless, they respectively suffer from a lack of precision and high costs. Building height can also be obtained via remote sensing technique, such as Synthetic Aperture Radar (SAR) (Li et al., 2020; Frantz et al., 2021), or take advantage of shadow in remote sensing images (Comber et al., 2011; Shao et al., 2011). However, in addition to the lack of precision, the absence of data necessary for modelling and the crowded character of rural buildings in China make the above methods difficult to be implemented. Compared to methods above, acquiring building height through UAV ensures high accuracy while being relatively efficient, and the method is quite simple, which also reduces the required costs.

These paragraphs have been added to the Introduction section of the revised manuscript.

**Specific modifications:**

**(1)** In Line 52-53, several typhoons have caused disasters, this sentence may need a reference.

**[Reply]**

Thanks for your comment. Two references have been added, and the sentence has been revised to reflect that "For example, Typhoon Hato in 2017, Typhoon Mangkhut in 2018, Typhoon Lekima in 2019 has caused significant damage to coastal cities in China, and resulted great losses of life and property (Zhou et al., 2021; Yang et al., 2019).".

**(2)** In Line 63, 'However' seems unnecessary, consider deleting it.

**[Reply]**

Thanks for your comment. The word has been removed in the manuscript.

**(3)** In Line 74-76, the sentence has some grammatical errors.

**[Reply]**

Thanks for your correction. The sentence has been removed in the manuscript due to content adjustments.

**(4)** In Line 81, 's' should be added to 'result'.

**[Reply]**

Thanks for your correction. The sentence has been revised based on your feedback.

**(5)** In Line 170-171, the use of 'such as' is suspiciously ambiguous, consider modifying the sentence.

**[Reply]**

Thanks for your comment. The sentence has been revised to reflect that "However, just as mentioned above, Guangdong is relatively vulnerable to storm surges, such as Typhoon Hato and Typhoon Mangkhut, due to its geographical characteristics.".

**(6)** In Fig.1, some of the color in Fig.1(c) are too similar, which might cause difficulty in reading, consider modifying it.

**[Reply]**

Thanks for your comment. Figure 1 has been replaced by followed image:

[Figure]

**(7)** In Line 218, according to what follows, it should be retitled wind field data, and the description of 'Holland' is needed.

**[Reply]**

Thanks for your comment. The paragraph has been reconstructed in the manuscript as follows:

(5) Hybrid wind field: ERA5 is the fifth generation of the European Reanalysis dataset produced by the European Centre for Medium-Range Weather Forecasts (ECMWF), and it provides the comprehensive and high-resolution atmospheric and climate data. Holland typhoon wind field model was proposed by Holland in 1980, which introduced Holland B parameter on the basis of the Schloemer exponential pressure distribution model (Holland, 1980). In this study, the two data are fused to generate hybrid wind field data, which is subsequently utilized for storm surge simulations.

**(8)** In Line 361, the sentence uses 'more', but there is no object of comparison, consider modifying it.

**[Reply]**

Thanks for your comment. The word has been removed in the sentence.

**(9)** In Line 529, 'Fig.9' should be 'Fig.9(b,c)'.

**[Reply]**

Thanks for your correction. The sentence has been revised based on your feedback.

**(10)** In Line 545, 'Fig.9' should be 'Fig.9(d,e)'.

**[Reply]**

Thanks for your correction. The sentence has been revised based on your feedback.

**(11)** In Line 555, while 'risk matrix' is introduced in this paragraph, there is no description of what exactly the 'risk matrix' is. It might be helpful to attach a table.

**[Reply]**

Thanks for your suggestion. The concrete representation of the risk matrix used for qualitative risk assessment is shown as follow:

Table 4. The concrete representation of the risk matrix

| | | **Vulnerability** | | | |
|---|---|---|---|---|---|
| | | Low | Moderate | High | Very High |
| **Hazard** | Low | Low | Low | Moderate | Moderate |
| | Moderate | Low | Moderate | High | High |
| | High | Moderate | High | High | Very High |
| | Very High | Moderate | High | Very High | Very High |

The table and necessary statements have been added into the section 4.3 of the revised manuscript.

**Reference:**

Comber, A., Umezaki, M., Zhou, R., Ding, Y., Li, Y., Fu, H., Jiang, H., and Tewkesbury, A.: Using shadows in high-resolution imagery to determine building height, Remote Sensing Letters, 3, 551-556, 10.1080/01431161.2011.635161, 2011.

Frantz, D., Schug, F., Okujeni, A., Navacchi, C., Wagner, W., van der Linden, S., and Hostert, P.: National-scale mapping of building height using Sentinel-1 and Sentinel-2 time series, Remote Sens Environ, 252, 112128, 10.1016/j.rse.2020.112128, 2021.

Hasanzadeh Nafari, R., Ngo, T., and Lehman, W.: Calibration and validation of FLFArs -- a new flood loss function for Australian residential structures, Natural Hazards and Earth System Sciences, 16, 15-27, 10.5194/nhess-16-15-2016, 2016.

Holland, G. J.: An analytic model of the wind and pressure profiles in hurricanes, 1980.

Huang, H., Chen, P., Xu, X., Liu, C., Wang, J., Liu, C., Clinton, N., and Gong, P.: Estimating building height in China from ALOS AW3D30, ISPRS Journal of Photogrammetry and Remote Sensing, 185, 146-157, 10.1016/j.isprsjprs.2022.01.022, 2022.

Li, X., Zhou, Y., Gong, P., Seto, K. C., and Clinton, N.: Developing a method to estimate building height from Sentinel-1 data, Remote Sensing of Environment, 240, 10.1016/j.rse.2020.111705, 2020.

Marvi, M. T.: A review of flood damage analysis for a building structure and contents, Natural Hazards, 102, 967-995, 10.1007/s11069-020-03941-w, 2020.

Shao, Y., Taff, G. N., and Walsh, S. J.: Shadow detection and building-height estimation using IKONOS data, International Journal of Remote Sensing, 32, 6929-6944, 10.1080/01431161.2010.517226, 2011.

Taramelli, A., Righini, M., Valentini, E., Alfieri, L., Gatti, I., and Gabellani, S.: Building-scale flood loss estimation through vulnerability pattern characterization: application to an urban flood in Milan, Italy, Natural Hazards and Earth System Sciences, 22, 3543-3569, 10.5194/nhess-22-3543-2022, 2022.

Thieken, A. H., Olschewski, A., Kreibich, H., Kobsch, S., and Merz, B.: Development and evaluation of FLEMOps – a new Flood Loss Estimation MOdel for the private sector, Flood Recovery, Innovation and Response I, 10.2495/friar080301, 2008.

Yang, J., Li, L., Zhao, K., Wang, P., Wang, D., Sou, I. M., Yang, Z., Hu, J., Tang, X., Mok, K. M., and Liu, P. L. F.: A Comparative Study of Typhoon Hato (2017) and Typhoon Mangkhut (2018)—Their Impacts on Coastal Inundation in Macau, Journal of Geophysical Research: Oceans, 124, 9590-9619, 10.1029/2019jc015249, 2019.

Zhou, C., Chen, P., Yang, S., Zheng, F., Yu, H., Tang, J., Lu, Y., Chen, G., Lu, X., Zhang, X., and Sun, J.: The impact of Typhoon Lekima (2019) on East China: a postevent survey in Wenzhou City and Taizhou City, Frontiers of Earth Science, 16, 109-120, 10.1007/s11707-020-0856-7, 2021.